**communications** engineering

# Cryogenic quantum computer control signal generation using high-electron-mobility transistors
Alberto Ferraris [1,2] ✉, Eunjung Cha[1], Peter Mueller [1], Kirsten Moselund[2,3] & Cezar B. Zota[1]

Multiplexed local charge storage, close to quantum processors at cryogenic temperatures could generate a multitude of control signals, for electronics or qubits, in an efficient manner. Such cryogenic electronics require generating quasi-static control signals with small area footprint, low noise, high stability, low power dissipation and, ideally, in a multiplexed fashion to reduce the number of input/ outputs. In this work, we integrate capacitors with cryogenic high-electron mobility transistor (HEMT) arrays and demonstrate quasi-static bias generation using gate pulses controlled in time and frequency domains. Multi-channel bias generation is also demonstrated. Operation at 4 K exhibits improved bias signal variability and greatly reduced subthreshold swing, reaching values of ~6 mV/ decade. Due to the very low threshold voltage of 80 mV at 4 K and the steep subthreshold swing, these circuits can provide an advantage over the silicon-based complementary metal-oxide-semiconductor equivalents by allowing operation at significantly reduced drive bias in the low output voltage regime <1 V. Together with their high-speed operation, this makes HEMTs an attractive platform for future cryogenic signal generation electronics in quantum computers.

In the last years, multiple advanced implementations of quantum computers have been realized, which promise to solve problems that are currently intractable with classical computers[1–4]. To facilitate the next generation of quantum computers[5], the development of more advanced classical control and readout electronics may be beneficial[6–10]. Such electronics should enable the use of fewer connections between the different temperature stages of the cryostat[11] and use fewer, less complex and/or more energy-efficient circuits to control and read qubits, for example through multiplexing[9,12]. Integration of key electronic components inside the cryostat, i.e. cryogenic electronics, to alleviate some of these constraints, is for these reasons now receiving increasing research attention[13–16].

A method to improve DC biasing of quantum computers along these lines is to use circuit cells formed by one transistor and one capacitor, able to store fixed charges similar to a dynamic random access memory (DRAM)[17,18] but with multiple controllable levels, that can be placed close to the qubits in the cryogenic environment. Implementations of this idea have been explored in literature in various platforms[19–21]. An exciting opportunity is to demonstrate cryogenic biasing electronics that leverage advantageous changes in device performance at cryogenic temperature[22–25]. While standard Si CMOS exhibits steeper switching characteristics, i.e., lower subthreshold swing, and increased drive currents due to higher carrier mobilities, these beneficial effects are limited

due to surface and defect scattering in surface-channel transistors[23,24]. Towards this end, other device platforms than Si CMOS[26–29], such as quantum well transistors, with even more beneficial temperature-dependent properties, are interesting[30–32].

In this work, we present an implementation of cryogenic capacitor-based charge storage, together with a method of operation that allows to utilize a single external voltage supply to charge multiple capacitors and hold unique deterministic voltages across them. We use high-electron mobility transistors (HEMTs) for these circuits, which exhibit sharply improved on- and off-state performance at cryogenic temperatures due to the large increase of electron mobility and buried channel operation[33]. This device technology is similar to that used in cryogenic low-noise amplifiers employed in qubit readout[34]. In particular, we demonstrate normally-off HEMT transistors that can be turned on with a threshold voltage, $V_T$, as low as ~80 mV at 4 K, which is important to reduce operating voltages and charge injection[33]. Our results indicate that this is a promising approach for generating qubit and electronics control signals at cryogenic temperatures in a scalable manner.

A scheme of the semiconductor heterostructure stack used in the fabricated HEMTs together with the integrated capacitor is shown in Fig. 1a (for a more detailed description of the fabrication process, see the Methods section). For this architecture, an array of cells is envisioned that is either

[1]IBM Research Europe – Zürich, Rüschlikon, Switzerland. [2]EPFL – Ecole Polytechnique Fédérale de Lausanne, Lausanne, Switzerland. [3]Paul Scherrer Institut, LNQ, Villigen, Switzerland. ✉e-mail: rra@zurich.ibm.com

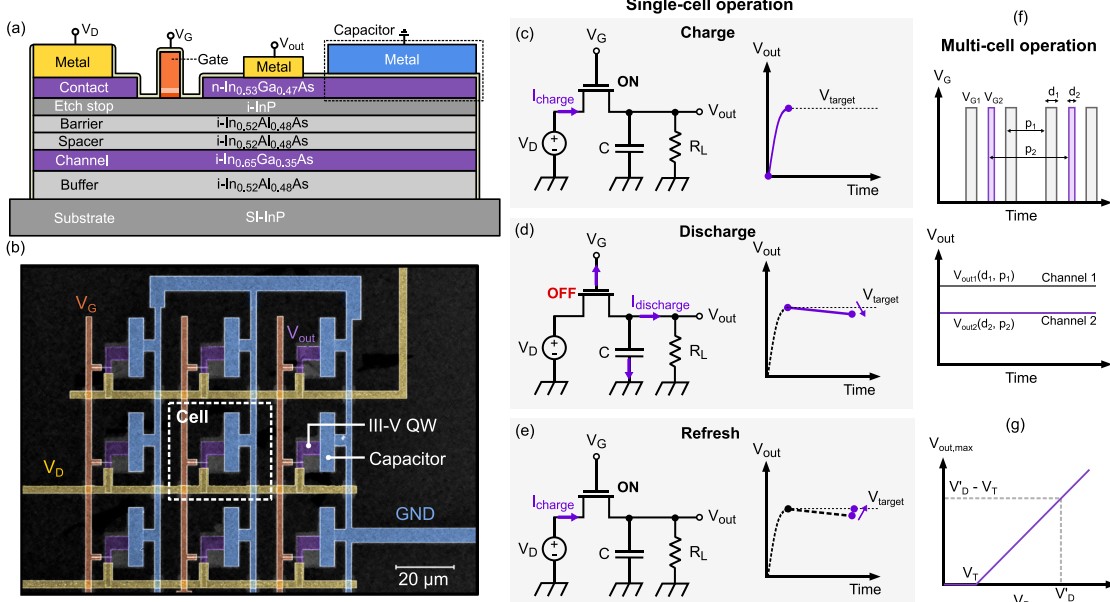

**Fig. 1 | Device concept and circuit operation. a** Schematic representation of the cross-section of a charge storage cell, showing a high-electron-mobility transistor (HEMT) and a parallel plate capacitor integrated in the same heterostructure. **b** Colored scanning electron microscope (SEM) image of a 3×3 array of charge storage cells. The different colors highlight the metal lines carrying the gate voltage ($V_G$, orange), drain voltage ($V_D$, yellow) signals and ground (GND, light blue); as well as the III-V semiconductor quantum well heterostructure (III-V QW, purple) that forms the transistor body. **c–e** Schematic figures representing the operation principles of the charge storage cell. The first step consists in turning on the transistor in order to charge the capacitor up to the desired voltage (which depends on

the charging time), then the transistor is turned off and the capacitor slowly discharges due to small leakage currents. After a certain time the transistor is briefly turned on in order to refresh the voltage dropping on the capacitor. **f** Schematic figures representing the operation of two cells in parallel, controlled by two different gate voltages $V_{G1}$ and $V_{G2}$, that produce pulses of different duration $d$ and period $p$, the result are two different almost-constant voltages at the output called $V_{out1}$ and $V_{out2}$ whose voltage level will depend on $d$ and $p$ of the respective gate signal. **g** Maximum output voltage that can be obtained on a cell as a function of the applied drain voltage $V_D$.

placed on the same substrate as, e.g. a quantum processor, or stacked vertically on top and connected using vias. Towards such a system, heterogenous integration approaches combining qubit and electronics layers will be required, or approaches leveraging highly modular sub-systems, similar to chiplets technologies.

In Fig. 1b, we show a realized 3×3 charge storage array, where each cell has a size of $(30 \times 30)\ \mu m^2$. Ideally, the charge storage cells should have a small pitch to improve density, for this reason we have scaled the size of the HEMT transistors down to a channel width of 1.3 μm. Most of the area of the array is occupied by the metal-oxide-semiconductor capacitors, that are realized with a parallel plate layout, and by the metal lines used to route the signals. The area occupied by the capacitors could be reduced significantly with a vertical fabrication process like the one used for standard DRAM memories[35], while the space occupied by the metal lines can be reduced by using vias to place the metal lines on a different layer with respect to the transistors and capacitors. The size of the capacitors can be reduced, e.g., by increasing the frequency of the cell charging operation.

Similarly to a DRAM matrix, the array is controlled by word and bit supply lines, that activate a specific row and column, to select one target cell. This turns on the transistor of the selected cell, charging the capacitor connected to it, as shown in Fig. 1c. In this study, we assume the circuit to be biasing high impedance loads, such as a qubit gate, that act as loads $R_L$ for each charge storage cell. The capacitor of each cell will then work as a voltage source, e.g., for the connected gate, and will discharge slowly due to small leakage currents in the transistor, capacitor and control input/gate. This process is illustrated in Fig. 1d. In this way, an external controller is still needed to bias the qubits, but it is possible to reduce the number of connections needed to access each DC control line from $M$ to $2\sqrt{M}$[21], and the external lines can be controlled by binary signals that change with time instead of by a DAC to directly generate voltage levels.

Due to the positive $V_T$ at cryogenic temperatures, the transistors are normally off and can be turned on sequentially for a fixed time interval, in order to charge the capacitor from the voltage supply. However, it is also possible to send a short pulse on the gate of the transistor, turning it on for a limited amount of time, which means that the capacitor will charge only partially with a final voltage that depends exponentially on the RC charging time. The gate pulse time in the charging operation can thus be tuned in order to obtain a voltage that is different from the original voltage supply and that can be different in different cells. The operation of multiple cells in parallel is illustrated in Fig. 1f. In the top panel, two trains of pulses of lengths $d_1$ and $d_2$ and periods $p_1$ and $p_2$ are applied on two charge storage cells. No specific waveforms are required for the gate pulses. In the following experiments, we use simple pulse trains at determined voltage levels and with indicated pulse durations applied to the transistor gates. In the bottom panel, the resulting output voltages are represented, which are two quasi-constant and simultaneous voltages whose values depend on the pulse length and frequency of the gate pulses applied to the respective cell.

With this circuit, the maximum voltage that can drop on the capacitor, $V_{out,max}$, is equal to the bias voltage minus the threshold voltage of the transistor ($V_D - V_T$), as shown in Fig. 1(g). The required precision of such a stored voltage for quantum applications is likely in the μV to mV range, depending on the control line type, coupling and function[17,20]. This operation mode is similar to charge-locking using sample-and-hold. Variability of $V_T$ at the transistor level should be kept low, since it will impact the specific stored voltage at a given bias level, though this can be adjusted for after a simple calibration step. The main sources of error in the voltage held by the capacitor are the thermal noise of the capacitor, small voltage offsets due to the transistor charge injection, and the voltage partition between the transistor gate

capacitance and the holding capacitance. The expressions for these error sources are the following[20]:

- Thermal noise: $V_{\text{rms}} = \sqrt{k_B T / C_H}$ where $C_H$ is the capacitance value of the holding capacitor, $k_B$ is the Boltzmann constant and $T$ is the temperature.
- Charge injection offset:

$$\Delta V_C = \frac{C_g}{2 C_H}(V_G^{\text{ON}} - V_D - V_T)$$

where $C_g$ is the gate capacitance of the transistor, $V_G^{\text{ON}}$ is the gate voltage used to keep the transistor in the on state, $V_D$ is the voltage applied to the drain of the transistor by the external voltage supply and $V_T$ is the threshold voltage of the transistor.

- Voltage partition offset:

$$\Delta V_P = \left(V_G^{\text{ON}} - V_G^{\text{OFF}}\right) \frac{C_{GS}}{C_{GS} + C_H}$$

where $V_G^{\text{OFF}}$ is the gate voltage in the off state, and $C_{GS}$ is the gate-to-source capacitance of the transistor.

All these error contributions can be mitigated by using a larger capacitor, a shorter transistor channel width and by applying lower voltages on the transistor gate, which is possible if the transistor has a low $V_T$. However, there are tradeoffs in the choice of each design parameter, as a large capacitor will increase the footprint and a transistor with a smaller width will make charging slower. Using a transistor with a smaller on-resistance, typically due to higher carrier mobility, such as in a HEMT, can in this regard improve the scalability of the cell.

Moreover, the charge held by the capacitor will slowly dissipate, due to the finite impedance of the transistor in the off state. In order to guarantee a stable voltage at the output, it is possible to periodically send refresh pulses, that turn the transistor of a cell on for a short period of time, allowing to compensate for the slow discharge of the capacitor.

## Results

### HEMT performance at cryogenic temperatures

Fig. 2a shows the transfer characteristics of a fabricated HEMT transistor from 300 to 2 K ambient temperature. The performance of the transistor improves at cryogenic temperatures, showing an on-off ratio of ~10^6, an off-state current as low as $2\times10^{-10}$ A/µm and a steeper transition, i.e. sub-threshold swing, between the on- and the off-state. The drain voltage, $V_D$, is here 0.1 V. At the transistor level, $V_D$ will influence the output current and to a lesser degree the subthreshold swing. At the circuit level, $V_D$ determines the maximum possible stored voltage on the capacitor. The subthreshold swing is shown in Fig. 2b and exhibits values as low as ~6 mV/decade, and stays below 20 mV/decade for more than four decades of drain current. In

Fig. 2c, the threshold voltage of a HEMT is extracted from the transfer characteristics using the linear extrapolation method, in which a linear equation is fitted to the $I_D$-$V_G$ characteristics at the point of its maximum slope[36]. The threshold voltage is negative at room temperature and increases at cryogenic temperatures, reaching ~80 mV at 2 K. While an increase in $V_T$ at lower temperatures is common in FET transistors, most HEMTs show a negative $V_T$ from 300 to 2 K to align the peak transconductance gain with 0 V gate voltage for high-frequency amplification. In this case, a slightly positive $V_T$ is instead desired because the transistors are used as logic switches, rather than high-frequency amplifiers, and stay in the off-state for most of the time. The threshold voltage was here engineered using a gate sinking process, whereby a thin layer of Pt is annealed into the gate barrier (see Methods).

### Charge storage functionality

The charge and discharge operations were characterized using a train of voltage pulses, $V_G$, with variable frequency $f = 1/p$, and duration $d$, applied to the gate of a transistor (Fig. 3a–d). The resulting voltage obtained at the output $V_{\text{out}}$, varies as a function of $f$ and $d$. In Fig. 3e, a gate charging pulse with $d = 50$ ns is shown in black as a function of time, and the other curves represent the resulting $V_{\text{out}}$ measured just before and after the pulse, for different values of $f$, meaning that the time period between pulses, $p$, is changed. As expected, the voltage at the output increases when the pulses are more frequent since there is less discharge between two charging events. This can be seen more in detail in Fig. 3a, where the output voltage level reached after a charging pulse is plotted as a function of $p$ and $f$ at a temperature of 5 K. By tuning both length and frequency of the pulses the whole voltage interval between 0 V and the maximum value of $\left(V_G - V_T\right) \sim 0.59 V$ can be covered at the output. However, not all the combinations of $d$ and $f$ are suitable for stable output voltage generation: as seen in Fig. 3e, the trains of pulses with lower frequency do not present a stable voltage at the output, because in the amount of time between two pulses the capacitor discharges significantly. This effect is unwanted as it represents a deviation from the desired quasi-static DC condition and should be minimized. To do so, in Fig. 3b, we show the variability of the output voltage, defined as the difference between the average voltage just after and just before a charging event. Reducing the charging pulse length reduces the variability, as less charge is injected in the capacitor, causing a smaller change in the voltage. Increasing the pulse frequency also reduces the variability, since the capacitor discharges less between two pulses. Figure 3a, b was obtained at room temperature. Since the transistors are normally on in this condition, a negative voltage was applied to the gate when a pulse was not present, to maintain the off state.

Figure 3c, d shows the same analysis at 2 K ambient temperature. In this case, the gate terminal is grounded when the transistor is in the off state, since the $V_T$ is positive. As observed, at cryogenic temperatures the generated output voltage is higher due to the reduced leakage currents, which

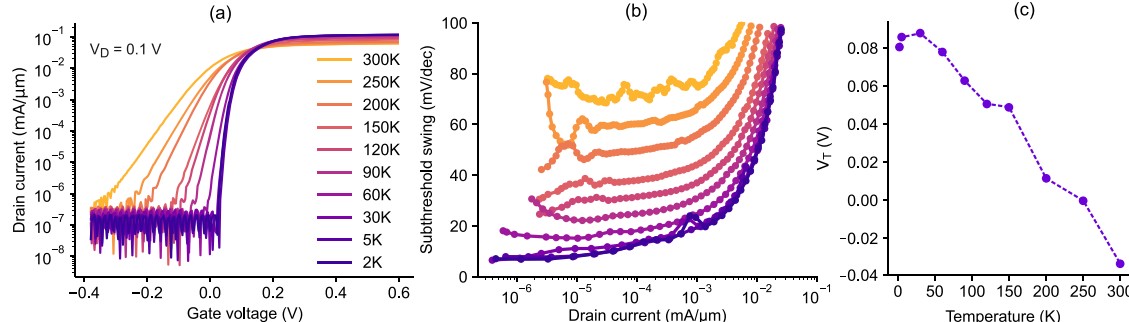

**Fig. 2 | Transistor characteristics at cryogenic temperatures. a** Transfer characteristics of a fabricated high-electron-mobility transistor (HEMT) measured between 300 K and 2 K, measured at $V_D = 0.1$ V. **b** Subthreshold swing of a HEMT transistor as a function of the drain current $I_D$ at different temperatures. A minimum value of 6 mV/decade is observed. **c** Threshold voltage of a HEMT as a function of the ambient temperature, which transitions from slightly negative at room temperature, to slightly positive at 2 K.

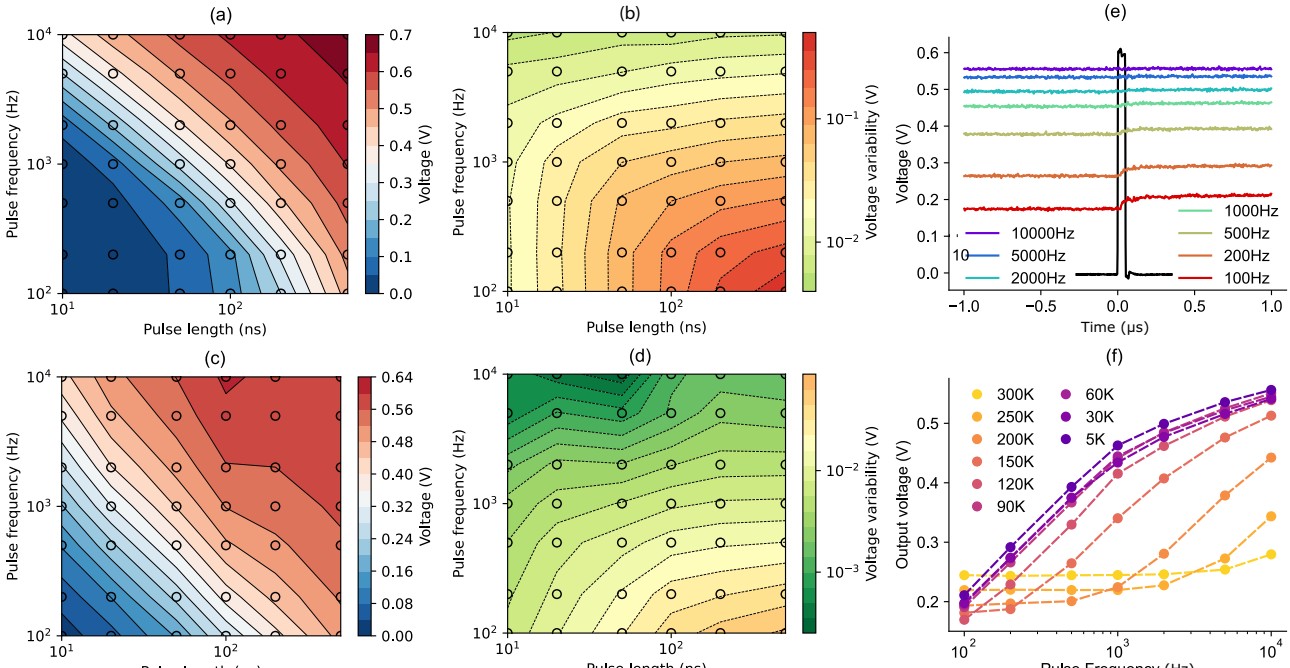

**Fig. 3 | Characterization of the output voltage and stability of a single cell.**
**a** Measurement of the output voltage as a function of the charging pulse length and frequency at $T = 300$ K. A negative voltage was applied on the gate when not charging in order to keep the transistor in the off state. The black circles highlight the points where the measurements were taken. **b** Measurement of the variability in the output voltage (obtained as the difference between the maximum and minimum value) as a function of the charging pulse length and frequency at $T = 300$ K. **c** Measurement of the output voltage as a function of the charging pulse length and frequency at $T = 2$ K. **d** Measurement of the variability in the output voltage as a function of the charging pulse length and frequency at $T = 2$ K. **e** Measurement of the output voltage in real time at $T = 5$ K. The charging pulse applied on the gate is displayed in black and its frequency is swept to obtain the other curves. **f** Comparison of the output voltage versus pulse frequency curves obtained at different temperatures. The voltage values are lower at higher temperatures due to higher leakages, and the transistors no longer turn completely off for temperatures above 150 K leading to a flattening of the output values.

translates into a reduced voltage variability. At 2 K, with $f = 10$ kHz and $d = 50$ ns, a voltage variation of 0.37 mV between the average voltage at the output just before and just after a charging pulse is shown.

A lower variability may be obtained by further increasing the frequency of the charging pulses, as the maximum $f$ tested here was 10 kHz. However, to maintain a fixed $V_{out}$ at increased $f$, the length of a pulse, $d$, must be reduced so that less charge is injected into the capacitor at each charging event. The HEMT transistors used for this device are capable of picosecond switching times, as their cutoff frequency is around 300 GHz, so the main limitation for achieving ultra-low variability is the capability of the control electronic circuits to generate pulses that are fast and short enough. Based on these considerations, we extrapolate from the trend in Fig. 3d that a pulse frequency of $f = 10$ MHz can result in a variability of <1 μV. To relax the constraint on the minimum pulse length it is possible to use transistors with larger on-state impedance, meaning that the charging time is increased at a given $d$. In this way, more frequent pulses need to be applied to obtain the same value of $V_{out}$, which will result in a lower variability. The disadvantage of this approach is that the maximum charging and discharging speed of the capacitor will be reduced.

Finally, in Fig. 3f, we focus on a single pulse shape and study the effect of temperature on the device response. As shown here, all voltage levels tend to decrease at higher temperatures. This is due to the higher leakage currents at higher temperatures that accelerate the discharge of the capacitor, and by the fact that the on-state impedance of a HEMT also increases at high temperature, thus reducing the amount of charge that is injected into the capacitor by a pulse of a given duration. Moreover, since the applied gate signal is the same at all temperatures, at above 150 K the transistors no longer turn off completely due to the $V_T$ shift, causing a current to leak from the drain preventing the generation of small voltages. This is a consequence of the optimization of the charge storage cells for cryogenic conditions.

The time constant associated with the charge of the capacitor at cryogenic temperature is $\tau_c \sim 10^{-6}$ s while the time constant associated with the discharge is $\tau_d \sim 10^{-1}$ s, e.g. for a qubit gate[21] (see Supplementary Fig. S1). This ratio does not depend on the value of the capacitor but on the ratios between the impedances of the leakage in the off state and the transistor in the on-state. The ratio of 5 orders of magnitude between the two gives us an estimation of the size of the cell arrays that could be controlled with this technique, assuming to address cells independently and one at a time.

## Operation with multiple channels

We next examine the operation of a 2 x 1 charge storage array (see Supplementary Fig. S2) to demonstrate independent control of each cell. In Fig. 4a, the two adjacent cells are controlled with gate pulses that have different frequencies at a temperature of 2 K. The pulses applied at the two transistor gates are shown in the top panel, while the bottom panel displays the resulting output voltages. As discussed previously, a higher pulse frequency, $f$, results in a higher output voltage, $V_{out}$, so that the first channel produces a higher output, $V_{out1}$, than the second, $V_{out2}$. This demonstrates that it is possible to tune the applied voltage on every cell independently only by using $f$, and without a need to tune the supply voltage.

In Fig. 4b, c, the crosstalk between the two cells is analysed, and in Fig. 4d, a circuit schematic of the electrical connections used for these measurements is shown. To characterize the crosstalk, the cells are controlled by pulse trains that are shifted by half a period, so that one of the two cells charges while the other discharges. Despite some variation between the two cells, likely caused by the fact that the device performance is in part determined by leakage currents that are not completely deterministic, in Fig. 4b, we do not observe an obvious correlation between the two cells. To investigate this further, in Fig. 4c a second measurement is performed to examine a smaller time scale around a charging pulse, ~10 μs for channel 2.

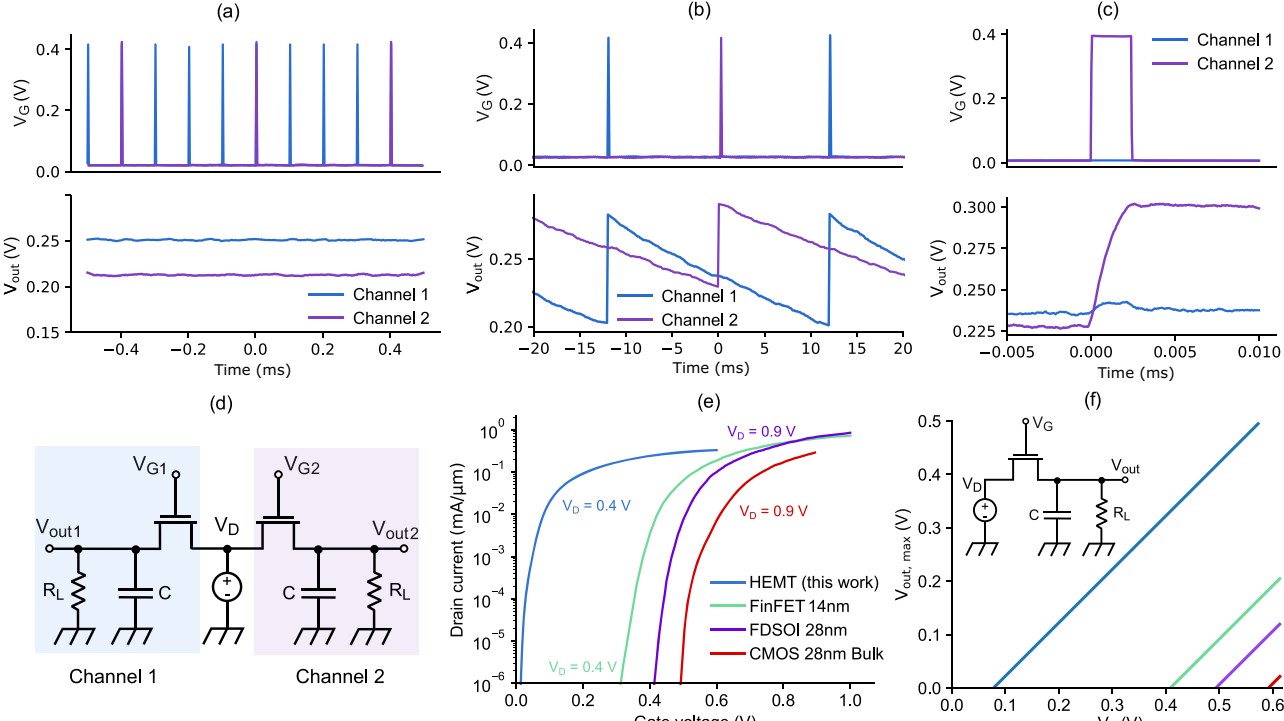

**Fig. 4 | Parallel operation of two cells and benchmarks. a** Top panel: gate voltage on two adjacent cells connected to the same drain voltage supply at $T = 2$ K; the frequency of the pulses is 10 kHz for channel 1 and 2.5 kHz for channel 2. Bottom panel: The resulting output voltages on the two channels. **b** Top panel: Gate voltage on two adjacent cells connected to the same drain voltage supply at $T = 2$ K; the frequency of the pulses is 50 Hz for both channels and the pulse length is 25 μs. Bottom panel: The resulting output voltages on the two channels. As shown, the influence of charging adjacent cells is minor. **c** Magnified version of Fig. 4b, with a focus on the central charging pulse. The voltage of channel 1 is temporarily slightly increased after channel 2 is charged. **d** The electrical scheme of the two adjacent cells that are

measured is presented. **e** Benchmark of transistor properties with respect to complementary metal-oxide semiconductor (CMOS). The high-electron-mobility transistor (HEMT) data was measured directly on the charge-storage devices presented in this paper, while the CMOS data is from the literature. The bias voltages and references are: HEMT (this work) $V_D = 0.4$ V; fin field-effect transistor (FinFET) 14 nm[25] $V_D = 0.4$ V; fully depleted silicon on insulator (FDSOI) 28 nm[23] $V_D = 0.9$ V; bulk CMOS 28 nm[24] $V_D = 0.9$ V. **f** Comparison of the maximum output voltage that can be achieved with respect to CMOS architectures. $V_T$ is computed with constant current method at the same current for all transistors.

In this case we observe an increase of ~6.5 mV on channel 1 during the charge that then stabilizes at ~2.1 mV above the original voltage value after the charging pulse has ended. These relatively high numbers are due to the long pulses used and the long distance in time between such pulses which cause an even greater variability of the output of the channel that is being charged. This analysis allows us to conclude that crosstalk between adjacent cells is a relatively small effect with respect to the variability of the output of a single channel (approximately one order of magnitude smaller) and that it should be considered important only when the applied voltage changes by a large value in a short amount of time.

**Transistor and charge storage properties with respect to CMOS**
The described biasing network was realized using HEMT technology, as it possesses a low SS combined with small positive $V_T$. This can be observed in Fig. 4e, where the transfer characteristics of a HEMT are compared to cryogenic Si CMOS technologies[23–25]. The HEMT transistor turns on at significantly lower gate voltages. This low $V_T$ translates into a wider operating window when the circuit is constrained by a fixed bias voltage $V_D$, as shown Fig. 4f. As described in Fig. 1g, the maximum output voltage that can be generated on the load is $V_D - V_T$ of the transistor. To expand this operating window, it is possible to uncouple the bias voltage for the gate and for the drain electrodes, thus allowing for a higher gate voltage while keeping the power consumption relatively low. This, however, means having two separate lines providing the bias voltage and adds complexity to the design of the circuits. Moreover, the steep turn-on of the HEMT, SS ~ 6 mV/dec., allows to reduce the charge injection[33], meaning that less charge is

accumulated or released from the channel when the transistor is switched on and off, with the benefit of reducing the noise transmitted to the qubit. A source of such charge may be parasitic capacitances in the clock path, which are expected to depend on the magnitude of the clock signal, i.e. $V_G$. In contrast, negative $V_T$ at room temperature (resulting in slightly positive cryogenic $V_T$) is not available in standard Si CMOS via metal work function engineering, though FDSOI CMOS may achieve this through the substrate bias. A typically observed saturation of SS around 20 mV/decade, as well as relatively modest mobility enhancement (10–30%) at low temperatures are further disadvantages of Si CMOS with respect to HEMTs[23,37]. Recently, some alternatives have been shown, like Si nanowire devices with ultra-steep cryogenic SS[38], that may be promising for similar applications, though sufficiently low $V_T$ must be engineered as well, which may be challenging due to energy quantization in 1D channels.

**Conclusions**
In this work, we demonstrate cryogenic circuits for the generation of multiple parallel DC control signal lines. This architecture, composed by charge storage cells, together with a frequency-based operation mode aims at reducing the number of signal lines needed to control the operation of a quantum processor.

InP/InGaAs HEMT technology is used for the implementation of these circuits, providing an on-off ratio of ~$10^6$, a minimum subthreshold swing of ~6 mV/decade, and a threshold voltage of ~80 mV at 4 K, which allows to operate the circuits at lower bias voltages than with standard cryogenic CMOS transistors. We demonstrate the ability to convert a set of periodic

pulses into a deterministic stable voltage generated by a cell at cryogenic temperature. We characterize the response of such cells to different parameters like temperature, pulse length and frequency, demonstrating the ability to tune the output voltage as needed. We also characterize the variability of the output voltage, and the crosstalk between two adjacent cells that are operated independently from each other. We conclude that the combination of attractive on- and off-state properties of the HEMTs, together with their high-speed operation, which enables charge storage with stable generated signal levels ( < 1 μV), make them attractive as a platform for future cryogenic DC signal generation electronics in quantum computers.

## Methods

### Device fabrication

The fabrication process is similar to that of our previous work[33]. Devices are fabricated by selectively etching an epitaxial heterostructure formed by a InGaAs, InAlAs and InP, by means of a wet etching process. Then, the source and drain contacts are realized by thermal evaporation of a Ni/Ge/Au metal stack that, when annealed partially diffuses in the heterostructure forming an ohmic connection with low resistivity. The gate of the transistor is realized first by etching a recess in the $n$-doped InGaAs cap layer and then by depositing a Pt/Ti/Pt/Au metal stack in the recessed region via evaporation. A 5 nm-thick $Al_2O_3$ passivation layer is deposited via atomic layer deposition (ALD) on top of the structure to protect the active regions of the transistors. During the ALD process the devices are exposed to heat, which has the effect of annealing the ohmic contacts and of allowing the diffusion of some Pt from the gate into the barrier layer, reducing the effective barrier thickness and allowing for a better electrostatic control on the channel[33]. Finally, the oxide is selectively etched, and an additional metal layer is deposited, creating the metal lines needed to route the signals and the measuring pads. This process allows to also fabricate the capacitors, without the need of additional steps. The last metal layer forms the top plate of the capacitors, the $Al_2O_3$ oxide serves as the dielectric and the $n$-doped InGaAs that is at the top of the heterostructure is the bottom electrode (i.e. a MOS capacitor).

### Electrical characterization

We used a Janis cryogenic probe station and a Quantum Design PPMS Dynacool to perform the measurements at cryogenic temperatures. The DC measurements were performed with a B1500 parameter analyser, while the measurement setup for the charge storage functionality consisted in an arbitrary waveform generator that generated the pulse signals, a DC voltage supply that supplied the fixed drain voltage and an oscilloscope to measure the voltage in real time. The 1 MΩ input impedance of the oscilloscope was found to be too low to properly characterize the device, since it caused all the charge of the capacitor to leak towards the measurement tool in a relatively short amount of time. To solve this issue a 100 MΩ resistor was placed in series to each of the oscilloscope channels. This increased the input impedance of the tool, thus increasing the discharge times in exchange for a reduction in the signal magnitude (voltage partition). Still, we expect a significant portion of the discharge current to be flowing in our measurement tool, meaning that if the tool is replaced with, for instance, a qubit gate with a significantly higher impedance, longer discharge times can be achieved. The voltage partition between the oscilloscope and the resistor caused a reduction in the SNR in the measurements since the signal gets reduced and a high resistance placed at room temperature is a source of thermal noise. To counter this effect, in Fig. 4a–c the high-frequency noise coming from the resistor was attenuated by applying a Savitzky–Golay filter to the measured data.

## Data availability

The data generated in this study is available on Zenodo[39].

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

## Acknowledgements

This work was supported as a part of NCCR SPIN, a National Centre of Competence in Research, funded by the Swiss National Science Foundation (grant number 51NF40-180604), by the European Union H2020 program SEQUENCE (Grant – 871764), by the SNF Ambizione and the BRNC.

## Author contributions

A. F. and C. Z. conceived and designed the devices. A. F. fabricated and characterized the devices, with the help of P. M (cryogenic measurements). A. F., C. Z., E. C. and K. M. discussed and interpreted the measurement data. A. F. wrote the manuscript with input from all authors.

## Competing interests

The authors declare no competing interests.
