## [Peer Review File · Communications Engineering]

RESPONSE TO REFEREES

Reviewer #1 (Remarks to the Author):

1. *The authors mentioned “For this architecture, an array of cells is envisioned that is either placed on the same substrate as, e.g. a quantum processor, or stacked vertically on top and connected using vias.” However, the intention of this sentence is quite confusing. How can we integrate this device together with the quantum process, which is typically a different platform?*

Reply: We thank the review for the question. As said in the cited sentence we imagine either to build transistors on the same substrate as the qubits (which will cause changes the transistor implementation) or to stack them vertically by bonding the HEMT substrate on top of the qubit one and connecting them through vias. It is true that qubit platforms are different than HEMT platforms. Towards this we may highlight the recent IBM qubit chips, such as the 433 qubit Heron, which is essentially a heterogenous system with qubit layers stacked on metal and Si layers (including a Si interposer). More information here: <https://www.ibm.com/quantum/blog/quantum-roadmap-2033>. Heterogenous integration in a quantum platform context is, in other words, already feasible, and could in the future incorporate biasing electronics to simplify wiring. We have added a statement to the manuscript to answer the reviewer’s question.

Changes: P2, L56: “For this architecture, an array of cells is envisioned that is either placed on the same substrate as, e.g. a quantum processor, or stacked vertically on top and connected using vias. Towards such a system, heterogenous integration approaches combining qubit and electronics layers will be required, or approaches leveraging highly modular sub-systems, similar to chiplets technologies.”

2. *The author needs to show the equivalent circuit diagram for the arrays. It is quite difficult to see what kinds of waveform or voltage will be distributed to each cell using their approach.*

Reply: We thank the reviewer for the comment. We refer to figure 4d, which has the circuit schematic for the two-cell circuit measured in this work, which should provide the answer for the reviewer’s request. Larger arrays can be scaled up from this prototype diagram. To answer the second part of the reviewer’s comments, we also clarified the text to better explain how the voltages and what waveforms are applied.

Changes: P3, L87: “Fig. 1(f). In the top panel, two trains of pulses of lengths d_1 and d_2 and periods p_1 and p_2 are applied on two charge storage cells. No specific waveforms are required for the gate pulses. In the following experiments, we use simple pulse trains at determined voltage levels and with indicated pulse durations applied to the transistor gates.”

3. *Fig. 2(a) should include V_D bias condition. Does this device show V_D dependence?*

Reply: We thank the reviewer for the suggestion. Similar to normal HEMTs the current in the device’s channel increases as the V_D is increased. This is shown in our previous works (Ferraris, IEDM 2022) and was not included in the manuscript as we do not wish to operate the device with varying values of V_D in this kind of application (in reality the V_s changes on its own during operation so the V_{ds} will also change causing an effect on the transistor; but we have taken this somehow into account by measuring the calibration curves directly). The ideal operation of the cell is to fix V_D and modulate the stored charge through the gate signal, as we demonstrate in this paper. In this case, V_D determines the maximum storage charge on the capacitor, which is shown in Fig. 4f.

Changes: Updated Figure 2(a) and its description to also include the V_d bias condition of the measurement. We also added the following statement to the manuscript: P5, L137 “The drain voltage, V_D , is here 0.1 V. At the transistor level, V_D will influence the output current and to a lesser degree the subthreshold swing. At the circuit level, V_D determines the maximum possible stored voltage on the capacitor.”

4. *Could the author make the statement “As seen in Fig. 3(e), some combinations do not present a stable voltage and show large variations over time in the output voltage due to the capacitor charging. “ much clearer? Which data are the authors mentioning?”*

Reply: We thank the reviewer for this question. We are here mentioning the curves obtained using the lower frequency trains of pulses shown in Figure 3(e), which show voltages that are not perfectly constant over time. This effect is unwanted and to solve it we have performed the measurements in Figures 4(a-d) that allow to find a good operating region for both length and frequency of the pulses, that produces the desired output voltage and a low variability. We have added a clarification about this to the manuscript text.

Changes: P 6, L. 157: “However, not all the combinations of d and f are suitable for stable output voltage generation: as seen in **Fig. 3(e)**, the trains of pulses with lower frequency do not present a stable voltage at the output, because in the amount of time between two pulses the capacitor discharges significantly.”

5. *As in Fig. 4(e), (f), the authors compared the IV and resulting V_{out} with other technologies. However, V_t seems to be adjusted even in other technologies. Then, authors should justify the benefits of HEMTs. Furthermore, there is a report showing 2.3 mV/dec in Si transistor at 5.5K (Han et al., VLSI symposium on Technology and Circuits 2023).*

Reply: We thank the reviewer for the question. The suggested reference shows excellent work on Si nanowire devices with ultra-low SS, which we have now added a reference to in the manuscript. However, it seems to have a V_T of 0.9 V for the n-type and -0.7 V for the p-type at 5 K so, despite the low SS, it is not well suited for low-power operation in this context. The authors are aware of the ongoing efforts in CMOS development to realize devices with lower V_T that are suitable for cryogenic low-power operation, and believe that if in the future a CMOS library with such characteristics is realized, it could be a viable alternative for the realization of the described circuit. However, at current writing, the combination of both ultra-steep SS and V_T close to 0 V appears to be unique for quantum well transistors, since SS in scaled standard Si CMOS tends to saturate around 15-20 mV/decade.

Changes: A new reference #39 was added. P. 8, L.245: “New developments exist, e. g., in Si nanowire devices, with ultra-steep cryogenic SS,³⁹ that may be promising for similar applications, though sufficiently low V_T must be engineered as well, which may be challenging due to energy quantization in 1D channels.”

6. *It seems that at the end, this device should be integrated CMOS, otherwise, signal input to V_{g1} , V_{g2} will not be available. Is there any strategy for that direction?*

Reply: We thank the reviewer for the question. The voltages controlling each cell must be generated by an external microcontroller. This does not need to be necessarily co-integrated with the charge storage cells, but can be connected using a set of external connections, similarly to what is currently done for the direct control of spin-qubits in a cryostat, where most of the electronics is actually at room temperature, outside of the fridge. We envision that in the future the controller could be cryogenic, but it could be placed at a different stage in the cryostat, placed at a higher temperature

which in return allows for an increased cooling power to offset the self-heating of the control electronics. We have added a statement to the manuscript to clarify this point.

Changes: P. 3, L. 88: “This process is illustrated in Fig. 1(d). In this way, an external controller is still needed to bias the qubits, but it is possible to reduce the number of connections needed to access each DC control line from M to $2\sqrt{M}$, and the external lines can be controlled by binary signals that change with time instead of by a DAC to directly generate voltage levels.”

7. *Device dimension and thickness details should be shown. ex) I cannot find the thickness of Al₂O₃ for the capacitor.*

Reply: We thank the reviewer for the comment, the oxide thickness was added in the description of the fabrication process.

Changes: P. 9, L. 278: “A 5 nm-thick Al₂O₃ passivation layer is deposited via atomic layer deposition (ALD) on top of the structure to protect the active regions of the transistors.”

8. *V_D should be shown in Fig. 4(e).*

Changes: Added the V_D values of each curve in the figure.

Reviewer #3 (Remarks to the Author):

1. *“The main sources of error in the voltage held by the capacitor are the thermal noise of the capacitor, small voltage offsets due to the transistor charge injection, and the voltage partition between the transistor gate capacitance and the holding capacitance.”*

In addition, the device-to-device process variation of HEMT can also cause significant errors. Authors should comment on the device variability of HEMT technology at cryogenic temperatures.

Reply: We thank the reviewer for this comment. It is true that device variability can have an impact on circuit operation as well, though probably not so much on the generated noise in quasi-static conditions. The most impactful variability is due to changes in V_T . Recent results have also shown that the variability of CMOS increases at cryogenic temperatures (Paz et al., VLSI, 2020). Analysis of 28 nm FDSOI showed that V_T variability is well described by Pelgrom’s law, meaning that variability depends inversely on the gate area, $LG \times WG$, where LG is the gate length, and WG is the gate width. However, in a HEMT, V_T does not so much depend on gate metal work function and surface charged defects, since the channel is relatively insulated from the surface. For that reason, there are good reasons to believe that variability may be lower than for Si CMOS. However, such large-scale studies don’t exist currently, since high-throughput cryogenic measurements are rather challenging to perform.

Changes: We have added a statement to clarify the role of variability: P.3, L96: “Variability of V_T at the transistor level should be kept low, since it will impact the specific stored voltage at a given bias level, though this can be adjusted for after a simple calibration step. The main sources of error in the voltage held by the capacitor are the thermal noise of the capacitor, small voltage offsets due to the transistor charge injection, and the voltage partition between the transistor gate capacitance and the holding capacitance.”

2. *“As seen before, all voltage levels tend to decrease at higher temperatures due to the higher leakage currents that accelerate the discharge of the capacitor.”*

Is it all because of the leakage though? Mobility also plays a significant role in determining the channel resistance and thus time constants. As the temperature increases, the mobility decreases, increasing the effective channel resistance and leading to larger RC time constants, which means for a given gate voltage, the $V_{out\ max}$ that the capacitor first charges to is smaller.

And, how does the capacitor discharge through the transistor’s channel leakage with the supply voltage V_d still connected during discharge as shown in Fig. 1(d)?

Reply: We thank the reviewer for the insightful comment. Regarding the theory of the temperature-dependence of the charge storage mechanism, we observe that the on-state resistance in a HEMT decreases by ~15% at cryogenic temperatures due to increased mobility and changes to the contact impedance, while the gate leakage can improve by one order of magnitude or more. In fact, the capacitor can discharge either through the load impedance (which can be the qubit gate or in our case a measurement tool) or towards the gate of the transistor, which during the “hold” or “discharge” state of Figure 1(d) is connected to ground in order to keep the transistor in the off state. HEMT transistors do not have a gate oxide like other FET technologies, and as a result they present a higher gate leakage at room temperature that significantly improves at cryo. As a result without the presence of an external load, the dominating effect should be the leakage reduction as we have described in the paper, since its change is more significant. However, the gate leakage is in parallel to the current

through the load impedance, which could be greater or smaller depending on the circuit configuration. In conclusion it is likely that both effects contribute to the observed result. We have added a section to the manuscript to clarify these points and the reviewer's comment.

Changes: P. 7, L. 192: "As shown here, all voltage levels tend to decrease at higher temperatures. This is due to the higher leakage currents at higher temperatures that accelerate the discharge of the capacitor, and by the fact that the on-state impedance of a HEMT also increases at high temperature, thus reducing the amount of charge that is injected into the capacitor by a pulse of a given duration."

3. *The thermal budget for peripheral circuitry in quantum computers is usually very limited. What are your comments on the power dissipation of the proposed scheme?*

Reply: We thank the reviewer for this question. While a single cell is holding a charge, the power dissipation will come mostly from the leakage. From the discharge time constant we can estimate the leakage current to be a fraction of a picoamp, which should be more than compatible with operation at cryogenic temperatures of multiple cells. A higher current will be generated if the voltage on the capacitor needs to be changed since a fast charge or discharge will generate a larger current in a small amount of time, the total power consumption would be increased depending on how often this happens.

4. *Mention the V_d used in Fig. 2(a).*

Reply: We thank the reviewer for the suggestion.

Changes: Updated Figure 2(a) and its description to also include the V_d bias condition of the measurement.

5. *Page 8, Line 229: V_{TH} should be V_T .*

Reply: Thanks.

Changes: Corrected.

6. *"In contrast is negative V_T at room temperature (resulting in slightly positive 238 cryogenic V_T) not available in standard Si CMOS via metal work function engineering," Check grammar.*

Changes: Corrected the sentence.